

# Otolith microchemistry and diadromy in Patagonian river fishes

Dominique Alò[1,2], Cristian Correa[3], Horacio Samaniego[2], Corey A. Krabbenhoft[4,5] and Thomas F. Turner[4]

[1] Departamento de Ecología, Pontificia Universidad Católica de Chile, Santiago, Chile
[2] Laboratorio de Ecoinformática, Instituto de Conservación Biodiversidad y Territorio, Universidad Austral de Chile, Valdivia, Los Ríos, Chile
[3] Instituto de Conservación Biodiversidad y Territorio, Universidad Austral de Chile, Valdivia, Los Ríos, Chile
[4] Department of Biology and Museum of Southwestern Biology, University of New Mexico, Albuquerque, NM, United States of America
[5] Department of Biological Sciences, Wayne State University, Detroit, MI, United States of America

## ABSTRACT

Coastal habitats in Chile are hypothesized to support a number of diadromous fish species. The objective of this study was to document migratory life histories of native galaxiids and introduced salmonids from a wide latitudinal range in Chilean Patagonia (39–48°S). Otolith microchemistry data were analysed using a recursive partitioning approach to test for diadromy. Based on annular analysis of Sr:Ca ratios, a diadromous life history was suggested for populations of native *Aplochiton taeniatus*, *A. marinus*, and *Galaxias maculatus*. Lifetime residency in freshwater was suggested for populations of *A. zebra* and *G. platei*. Among introduced salmonids, populations of *Oncorhynchus tshawytscha* and *O. kisutch* exhibited patterns consistent with anadromy, whereas the screened population of *O. mykiss* appeared restricted to freshwater. *Salmo trutta* exhibited variable patterns suggesting freshwater residency and possibly anadromy in one case. The capacity and geographic scope of hydropower development is increasing and may disrupt migratory routes of diadromous fishes. Identification of diadromous species is a critical first step for preventing their loss due to hydropower development.

## INTRODUCTION

Only 47 native and 27 non-native inland fish species are currently recognized in Chile, and several of these are thought to exhibit some tolerance for shifting between saline and freshwater habitats (*Dyer, 2000*; *Habit & Victoriano, 2005*; *Habit, Dyer & Vila, 2006*; *Vila et al., 2011*; *Ministerio del Medio Ambiente, 2013*; *Vargas, Arismendi & Gomez-Uchida, 2015*). Approximately 15% of these fishes are hypothesized to display diadromous migratory behaviour (Table S1), compared to less than 1% for fishes worldwide (*Nelson, 2006*).

The term diadromy describes regular, predictable, and physiologically mediated movements between freshwater and the sea. Diadromy necessitates profound physiological changes (i.e., osmoregulation) when shifting from marine to freshwater habitats and vice versa (*Gross, Coleman & McDowall, 1988*). Diadromy can be either obligatory or

Corresponding author
Dominique Alò,
dominiquealo@gmail.com

facultative (*Dingle & Drake, 2007*). The direction of migration depends on life history stages and habitats where reproductive and feeding events occur. The combination of these factors defines three different types of diadromy: anadromy, catadromy, and amphidromy (*Myers, 1949*; *Gross, 1987*; *Gross, Coleman & McDowall, 1988*; *McDowall, 1992*; *McDowall, 1997*; *Limburg et al., 2001*) (in particular refer to *McDowall (1997)* for a review of the terminology and a visual aid).

Given the high percentage of fishes in Chile hypothesized to exhibit some form of diadromy, migration might play an important, yet unrecognized role in establishing national priorities of aquatic biodiversity conservation. At present, a high percentage of the continental ichthyofauna in Chile is categorized with some degree of conservation threat by Chilean environmental agencies and other authors, although conservation categories can be incongruent and threats underestimated (*Habit & Victoriano, 2005*; *Diario Oficial de la Republica de Chile, 2008*; *Ministerio del Medio Ambiente, 2013*; *IUCN, 2015*; *Vargas, Arismendi & Gomez-Uchida, 2015*).

Coastal habitats in Chile appear well suited to support establishment of diadromous species. Andean rivers that flow into the Pacific Ocean include a variety of different habitats in a limited longitudinal distance (average 145 km), spanning from areas of rocky substrates, high elevation gradients, clear waters and low temperatures, to areas of low flow, sandy substrates, and aquatic vegetation (*Habit & Victoriano, 2005*; *Instituto Nacional de Estadisticas, 2015*). Spatial habitat heterogeneity is essential for maintenance and completion of diadromous life cycles, and for maintaining evolutionary potential (i.e., genetic diversity) for life history variation (*Pulido, 2007*; *Dingle, 2014*). Therefore, fragmentation events imposed by human-made barriers may affect fish fitness and restrict movement between habitats more so than in other areas (*Waples et al., 2007*).

Patagonian fishes offer a unique opportunity to understand migration patterns in relatively pristine habitats and establish a baseline against which future potential impacts associated with river impoundments can be compared. Despite strong economic growth and efforts to develop hydroelectric potential to meet the country's growing energetic demand (*Joo, Kim & Yoo, 2015*), many rivers in southern Chile are still free-flowing, offering opportunities to study pre-impoundment patterns of diadromous migration. In particular, galaxiid fishes are distributed across the temperate Southern Hemisphere (*McDowall, 2002b*) and diadromy seems to be a recurrent trait among many of the species (*McDowall, 1971*; *McDowall, 1988*). Likewise, salmonids are among some of the best studied diadromous fishes in the Northern Hemisphere and are now well established in southern Chile (*McDowall, 2002a*; *Correa & Gross, 2008*).

Using micro-geochemical data obtained from otoliths, this study sought to investigate whether native galaxiids and introduced salmonids exhibit diadromy in Chilean rivers. Otoliths are calcified deposits in the inner ear of fishes that accumulate in ring-like fashion over ontogenetic growth. Elemental analysis of otoliths can help to distinguish origins of marine and freshwater fishes among locations with variable water chemistry. Differing chemical composition of the otolith from the primordium (core) to the edge is indicative of the different environments in which a fish has lived and allows for hypothesis tests related to patterns of fish movement. When analysed sequentially across an otolith sagittal section,

changes in elemental ratios can inform fine-scale patterns of movement, connectivity, dispersal, and the location of natal habitats (*Halden et al., 2000*; *Howland et al., 2001*; *Kraus, 2004*; *Ashford et al., 2005*; *Campana, 2005*; *Arkhipkin, Schuchert & Danyushevsky, 2009*). To quantify these changes in Patagonian fishes, we applied univariate recursive partitioning approaches based on Classification and Regression Trees (CART) to detect discontinuities in elemental ratios that may indicate habitat shifts (*Vignon, 2015*).

This research was motivated by the necessity to understand the potential for movement in native and exotic fishes in non-impounded systems in southern Chile and used an existing sample pool (*Correa & Hendry, 2012*; *Correa, Bravo & Hendry, 2012*; *Alò et al., 2013*). Several native fishes have previously been hypothesised to exhibit capacity for shifting between marine and freshwater habitats (*Dyer, 2000*; *Habit & Victoriano, 2005*; *Habit, Dyer & Vila, 2006*; *Vila et al., 2011*; *Ministerio del Medio Ambiente, 2013*; *Górski et al., 2015*; *Vargas, Arismendi & Gomez-Uchida, 2015*) and some exotic salmonids appear to have established successful diadromous life-histories (*Ciancio et al., 2005*; *Ciancio et al., 2008*; *Riva-Rossi et al., 2007*; *Correa & Gross, 2008*; *Arismendi & Soto, 2012*; *Araya et al., 2014*; *Górski et al., 2016*). This study sought to characterize the potential diadromous characteristics from a wider diversity of native and exotic fish species in Patagonia, including previously unevaluated species sampled from a wider and unexplored geographic range. Inclusion of native and exotic species in this study provided a framework to compare well-known life-history characteristics of northern hemisphere salmonids to the lesser-known galaxiids.

## METHODS

### Fish collections

Between 2004 and 2011, specimens of *Aplochiton zebra, A. taeniatus, A. marinus, Galaxias maculatus, G. platei, Oncorynchus tshawytscha, O. kisutch, O. mykiss,* and *Salmo trutta* were collected using various methods from six locations across a large latitudinal range (39.5–48.0°S) in western Patagonia, Chile (Fig. 1, Table 1). At each location, fish specimens were euthanized by an overdose of anaesthetic solution (tricaine-methanesulfonate MS-222 or clove oil). Due to the difficulties in morphological identification, genetic data were used to identify individuals in the genus *Aplochiton* to the species level (*Alò et al., 2013*). The McGill University Animal Care Committee (UACC), Animal Use Protocol No. 5291, approved use and handling of animals.

### Otolith preparation

This study used the Strontium (Sr) to Calcium (Ca) molar ratio to infer habitat shifts across salinity gradients of Patagonian fishes. Strontium is particularly useful for reconstructing environmental history of fishes as it replaces Ca in the otolith matrix according to its availability in the fish habitat (*Secor & Rooker, 2000*; *Campana, 2005*; *Pracheil et al., 2014*).

Prior to specimen preservation, sagittal otoliths were extracted and either stored dry in test tubes or in 95% ethanol, as elemental compositions and structures of otoliths are not strongly affected by ethanol for the elements assayed (*Proctor & Thresher, 1998*). In the laboratory, otoliths were polished, cleaned, and mounted individually on clean glass slides
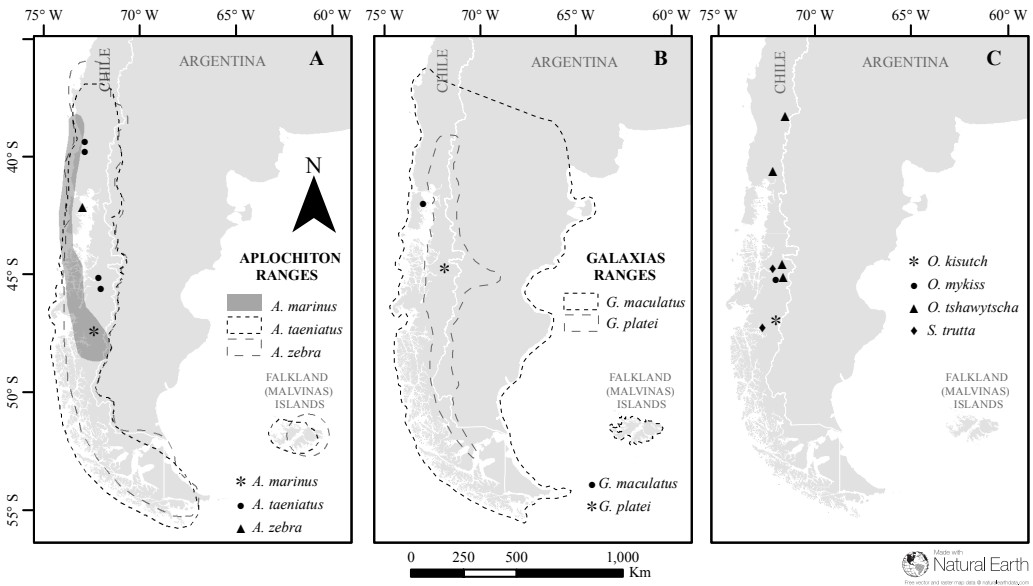

**Figure 1 Map of sampled species and locations across Patagonia.** Estimated distribution range for native Chilean galaxiids (shaded and dash lined polygons) and the sampling locations of specimens used in this study (dots) for (A) genus *Aplochiton*, (B) genus *Galaxias*, (C) non-native fishes examined in this study. Made with Natural Earth. Free vector and raster map data at https://naturalearthdata.com.

using a thermoplastic cement (Crystalbond™). In order to expose growth rings, 3M™ (fine) and Nanolap® Technologies (coarse) diamond lapping film wetted with deionized water was used to polish otoliths by hand until a satisfactory sagittal section of annuli was visible (*Fowler et al., 1995*). For *Aplochiton* spp., *Galaxias* spp., *S. trutta*, *O. kisutch* and *O. mykiss* otoliths, a 30-μm and then 3-μm lapping film was used to expose annuli and get a finished polish. *O. tshawytscha* otoliths required larger lapping film (45 and 60 μm) to reach an appropriate view, but were finished with 3 μm film for increased clarity. For larger otoliths (e.g., *O. tshawytscha*), it was sometimes necessary to polish on opposite sides to produce a thinner section.

Following polishing, the mounting adhesive was dissolved in a 100% acetone bath and sonicated for 10 min. Larger otoliths were cleaned a second time with acetone as needed. Each otolith was then sonicated twice in Milli-Q water for 5 to 10 min each. Following cleaning, otoliths were rinsed a final time in Milli-Q water, transferred to clean vials and placed in a positive laminar flow hood for 24–48 h to dry (similar to methods of *Elsdon & Gillanders (2002)*).

Acid-washed porcelain forceps were used to mount clean, dry otoliths on acid-washed microscope slides. Otoliths were grouped according to diameter and mounted 10–28 per slide accordingly. Each otolith was placed within one small drop of fresh Crystalbond melted onto a single side.

Slides were securely kept in acid-washed, sealed petri dishes for transport to Woods Hole Oceanographic Institute (Woods Hole, MA, USA). There, laser ablation-inductively coupled plasma-mass spectrometry (LA-ICP-MS) was conducted from October 8th to

Table 1 **Description of the samples studied and summary of results of migration pattern determination.** Values correspond to the percentage of individuals assigned to one of five possible patterns: freshwater resident (FW), brackish water resident (B), anadromous (ANA), catadromous (CAT), amphidromous (AMPH) or else omitted from interpretation due to uncertainties in the otolith transects (O). For individual results, see Table S2.

| | Species | N | Location | Year | Latitude | Longitude | FW | B | ANA | CAT | AMPH | O |
|---|---|---|---|---|---|---|---|---|---|---|---|---|
| | | | | | | | \multicolumn{6}{c}{Inferred Migratory Strategy (%)} | | | | | |
| Native Galaxiids | *A. zebra* (Jenyns, 1842) | 5 | Aysén-Caro | 2009 | −45.80 | −72.55 | 100 | | | | | |
| | *A. zebra* | 3 | Chiloé-Tocoihue | 2011 | −42.30 | −73.44 | | 100 | | | | |
| | *A. taeniatus* (Jenyns, 1842) | 8 | Valdivia-Santo Domingo | 2010 | −39.91 | −73.14 | | 12.5 | | 50 | 25 | 12.5 |
| | *A. taeniatus* | 2 | Valdivia-Lingue | 2011 | −39.46 | −73.09 | | | | 100 | | |
| | *A. taeniatus* | 3 | Aysén-Palos | 2007 | −45.32 | −72.70 | | | | 66.7 | | 33.3 |
| | *A. taeniatus* | 2 | Aysén-Caro | 2009 | −45.80 | −72.55 | 100 | | | | | |
| | *A. marinus* (Eigenmann, 1928) | 7 | Baker-estuary | 2007 | −47.79 | −73.52 | | | | 100 | | |
| | *G. platei* (Steindachner, 1898) | 2 | Aysén-Palos | 2007 | −45.32 | −72.70 | 100 | | | | | |
| | *G. maculatus* (Jenyns, 1842) | 4 | Chiloé-Tocoihue | 2011 | −42.30 | −73.44 | | 25 | | 50 | 25 | |
| Introduced Salmonids | *S. trutta* (Linnaeus, 1758) | 4 | Aysén-Palos | 2007 | −45.32 | −72.70 | 100 | | | | | |
| | *S. trutta* | 4 | Aysén-Caro | 2009 | −45.80 | −72.55 | 100 | | | | | |
| | *S. trutta* | 1 | Baker-estuary | 2007 | −47.79 | −73.52 | | | 100 | | | |
| | *O. tshawytscha* (Walbaum, 1792) | 8 | Aysén-Ñireguao | 2004 | −45.17 | −72.12 | | | 75 | | | 25 |
| | *O. tshawytscha* | 8 | Baker-Jaramillo | 2004 | −47.70 | −73.05 | 12.5 | | 50 | | | 37.5 |
| | *O. tshawytscha* | 6 | Petrohué-Patos | 2004 | −41.18 | −72.46 | | | 66.7 | | | 33.3 |
| | *O. tshawytscha* | 4 | Toltén-Peuco | 2004 | −38.85 | −71.76 | | | 50 | | | 50 |
| | *O. tshawytscha* | 4 | Toltén-Truful | 2004 | −38.85 | −71.67 | | | 50 | | | 50 |
| | *O. tshawytscha* | 5 | Aysén-Simpson | 2008 | −45.73 | −72.10 | 80 | | | | | 20 |
| | *O. kisutch* (Walbaum, 1792) | 2 | Baker-Vargas | 2007 | −47.68 | −73.04 | | | 100 | | | |
| | *O. mykiss* (Walbaum, 1792) | 3 | Aysén-Caro | 2009 | −45.80 | −72.55 | 100 | | | | | |

11th, 2012 (*Aplochiton* spp., *S. trutta*, *Galaxias* spp., *O. kisutch,* and *O. mykiss*) and again from February 4th to 5th, 2013 (*O. tshawytscha*). Laser ablation was performed with a large format laser ablation cell on a New Wave UP193 (Electro Scientific Industries, Portland, Oregon) short pulse width excimer laser ablation system. This was coupled with a Thermo Finnigan Element2 sector field argon plasma spectrometer (Thermo Electron Corporation, Bremen, Germany) for elemental analysis. The laser was configured for single pass, straight line scanning at a speed of 5 $\mu$m per second. The laser beam spot size was 50 $\mu$m at 75% intensity and 10 Hz pulse rate. Isotopic activity rates (counts per second) were determined for $^{87}$Sr and $^{47}$Ca. Certified standard FEBS-1 (*Sturgeon et al., 2005*) and a 1% nitric acid (HNO$_3$) blank were passed through the instrument before and after each block of 4 to 11 otoliths to minimize bias in elemental mass. Each otolith was visualized on screen and the intended ablation transect of each sample was plotted digitally and analysed by ablation with a laser beam (refer to Fig. 2 or supplemental pictures: https://doi.org/10.6084/m9.figshare.6387665.v2 for a visual example).

## Data analysis

Digital images of each otolith were taken after the laser ablation procedure with a digital Nikon Coolpix P6000 split with a Martin microscope MM Cool mounted on Nikon SMZ800 lenses and illuminated with a NII-LED High Intensity Illuminator (Supplemental pictures: https://doi.org/10.6084/m9.figshare.6387665.v2).

For accuracy, each raw data point was produced from an average of ten consecutive reads. Isotopic intensities were corrected by subtracting the means of the isotopic intensities detected in the blanks. Points below the limit of detection of the same isotope measured in the blanks (mean - 3SD) were eliminated. Isotopic intensities were converted to elemental intensities by taking into account the percent natural occurrence of the isotopes. Strontium was standardized to Ca and converted to molar Sr:Ca ratios using sequential measurements of the standard reference material and the atomic mass of the elements analysed (*Wells et al., 2003*; *Wolff, Johnson & Landress, 2013*) (raw data and Python scripts to convert from isotopic intensities to molar ratios provided as Dataset S1). Elemental data and otolith transects were rigorously checked, and outliers caused by recording errors were removed (additive outliers, R Package "tsoutliers" v. 0.6–5, (*López-de-Lacalle, 2016*)). The ideal double life-history transect obtained ran across each sagittal otolith and through the primordium, thus providing two similar (redundant) patterns related to life history variation, one on either side of the primordium. Interpretations were based on the analyses of both sides of each double transect, if possible. However, in a number of cases, transects were imperfect due to damaged otoliths or inaccurate ablation pathways (Supplemental pictures: https://doi.org/10.6084/m9.figshare.6387665.v2.) In these cases, data were analysed as partial transects that were used to differentiate putatively diadromous or resident signals (Fig. 2).

Classification and Regression Trees (CART, *Breiman et al., 1984*) were used to detect shifts in elemental ratios across the otolith transect. CART is an alternative to qualitative methods traditionally used to interpret the chronological signal in otolith microchemistry transects (*Vignon, 2015*). The position along the otolith transect (predictor variable) was

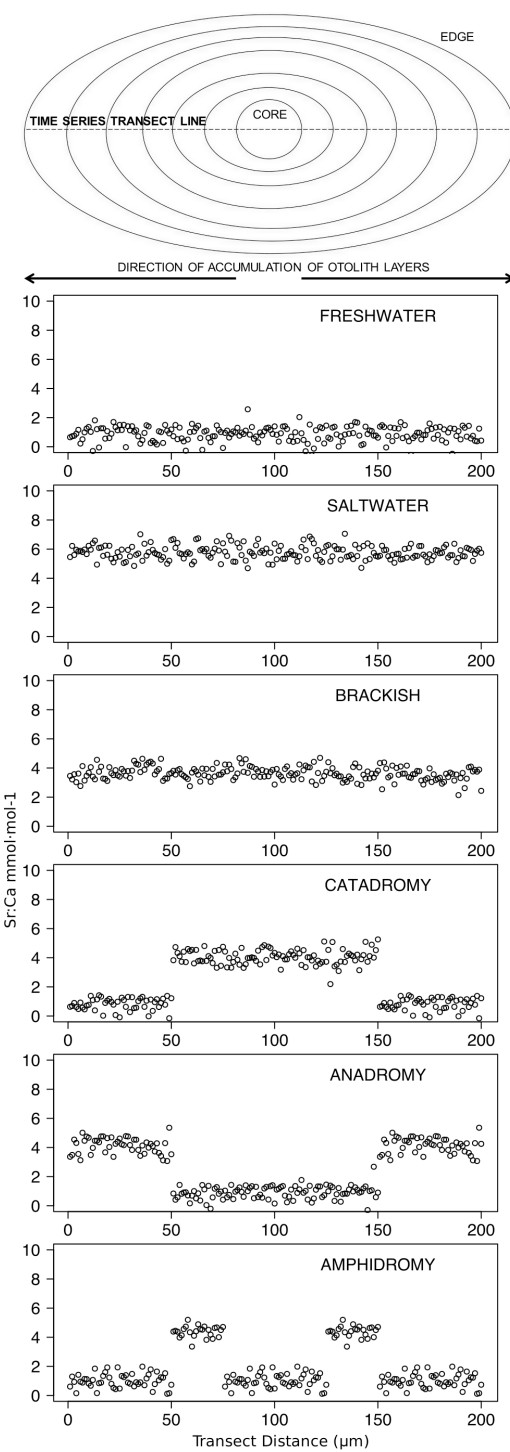

**Figure 2  Idealized representations of resident and diadromous life-histories as expected in otolith microchemistry results.** The uppermost image depicts a schematic representation of an otolith, showing how growth rings accrue over time around the core and culminate at the edge. The number of distinct layers in the otolith depends on the age of the individual. The images below represent idealized time series data obtained by repeatedly measuring (via laser ablation and spectrometry) elemental strontium to calcium (Sr:Ca) ratios across the otolith. Each box shows an expected time series for each life-history strategy.

recursively partitioned using regression trees in order to differentiate segments of the transect that shared similar mean Sr:Ca values (response variables) (*Breiman et al., 1984*; *Therneau & Atkinson, 1997*; *De'ath, 2002*; *Strobl, 2009*). CART was implemented in the Tampo library (version 1.0) for R statistical software 3.0.2 (*Vignon, 2015*).

Summary statistics of molar Sr:Ca ratios were calculated across all individuals. The main goal of this work was to represent movement patterns at a broad scale. Therefore, since shifts in Sr:Ca have been well documented in many species of fish moving across environments with variable Sr (*Tzeng & Tsai, 1994*; *Limburg, 1995*; *Tzeng, Severin & Wickstrom, 1997*; *Campana, 1999*; *Limburg et al., 2001*; *Chang, Iizuka & Tzeng, 2004*; *Araya et al., 2014*), CART analysis was used in a semi-supervised manner to identify the presence or absence of sudden discontinuities in the Sr:Ca otolith signal (*Vignon, 2015*). This was done by setting three progressively relaxed conditions to the splitting procedure. A splitting condition is the minimum difference in mean values between consecutive Sr:Ca profile segments beyond which the segments are permanently split in different categories. The three splitting conditions adopted were 0.5, 0.7, and 1.0. The detection of one or more discontinuities or splits in the Sr:Ca signal was interpreted as evidence in favour of diadromy, or otherwise, evidence in favour of residency. When diadromy was inferred, the direction of ontogenetic movements was deduced from differences in segment means; increasing values indicated movements towards the sea, and vice versa.

Further inference about habitat occupancy (freshwaters, estuaries, or the sea) required a visual, heuristic examination of Sr:Ca profiles in relation to (i) the sites of capture, (ii) comparisons with movement patterns inferred from all the species analysed, (iii) assumption of positive correlation between environmental Sr and marine influence (*Secor & Rooker, 2000*). Finally, all evidence was assembled to make individual inferences about migration patterns suggestive of amphidromy, catadromy, or anadromy. However, determining the precise extent of movements between habitats was beyond the scope of this work. Finally, otolith transect quality affected the confidence of our interpretations; from maximum confidence on inferences from transects that conformed to the hypotheses of the models proposed in Fig. 2, to uncertain interpretations from incomplete or faulty transects.

# RESULTS

Observed variation in Sr:Ca ratios (Fig. 3) reflected a wide gradient of average values (lowest mean ($\pm$sd) 0.66 ($\pm$0.088) in *G. platei* and highest mean 4.03 ($\pm$1.54) in *G. maculatus*). Depending on the species or population analysed, CART identified patterns of change in Sr:Ca elemental ratios consistent with different migratory life histories proposed in the schematic representation in Fig. 2. Representative individuals for each species are shown in Fig. 4. Details on the splitting results of all individuals under different stringency conditions are given in Fig. 5 whereas Table S2 reports details on the mean and standard deviation at each split. A summary of the inferred migratory strategy for each species is shown in Table 1.

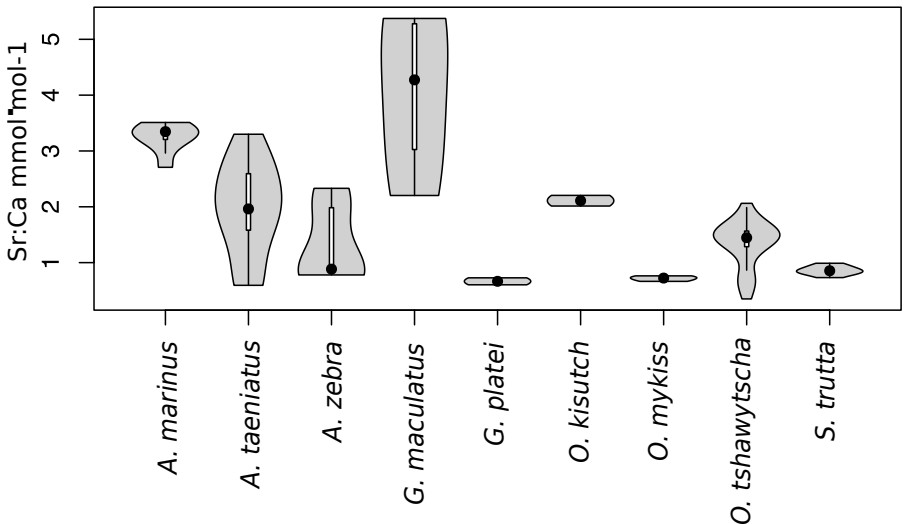

**Figure 3** **Violin plot for Sr:Ca values.** This plot reports the Sr:Ca values for otoliths grouped by species. Black dot, median; white line, first to third quartile. Grey areas, kernel density plot.

## Native galaxiids

Large elemental shifts in otolith profiles suggested a catadromous life-history for most *A. taeniatus* analysed (Table 1 and Fig. 5). Even when confined to strictly freshwater habitats, as in Lake Caro, *A. taeniatus* juveniles showed a mild shift in Sr:Ca suggesting movements between freshwater habitats (Fig. 5, cond. = 0.5), as contrasted with *A. zebra* or *O. mykiss*, which showed no such shifts.

Otolith profiles (Figs. 4 and 5) suggested that *A. marinus* copes with high levels of salinity variation in the Baker River system. Otolith primordia of all specimens of *A. marinus* showed evidence of higher Sr:Ca ratios at early stages of growth, presumably before the fish entered the estuary (site of capture). Taken together, these data suggest that *A. marinus* is catadromous.

Results indicated that *A. zebra* uses a chemically uniform habitat at both collection localities (Fig. 4), although results should be corroborated by future studies because *A. zebra* individuals assayed were juveniles. Nevertheless, specimens from Tocoihue River appear to have been exposed to marine influence compared to those from Lake Caro (Table S2), suggesting preference for freshwater residency, but capacity for osmoregulation when salinity levels increase.

*G. maculatus* individuals were sampled from the same site as some specimens of *A. zebra* (Tocoihue). However, among four specimens assayed, three showed the highest variation in Sr:Ca compared to any other galaxiid species analysed (Fig. 3) with evidence of both catadromous and amphidromous transitions (Table 1). The *G. maculatus* specimen in Fig. 4 was caught in the lower reach of the Tocoihue River in an area with strong tidal influence. The otolith profile suggests an amphidromous life-cycle with intermediate Sr:Ca levels in the primordium followed by increases in Sr:Ca ratio and subsequent decrease to lower Sr:Ca levels. A fourth specimen of *G. maculatus* showed no major Sr:Ca fluctuations

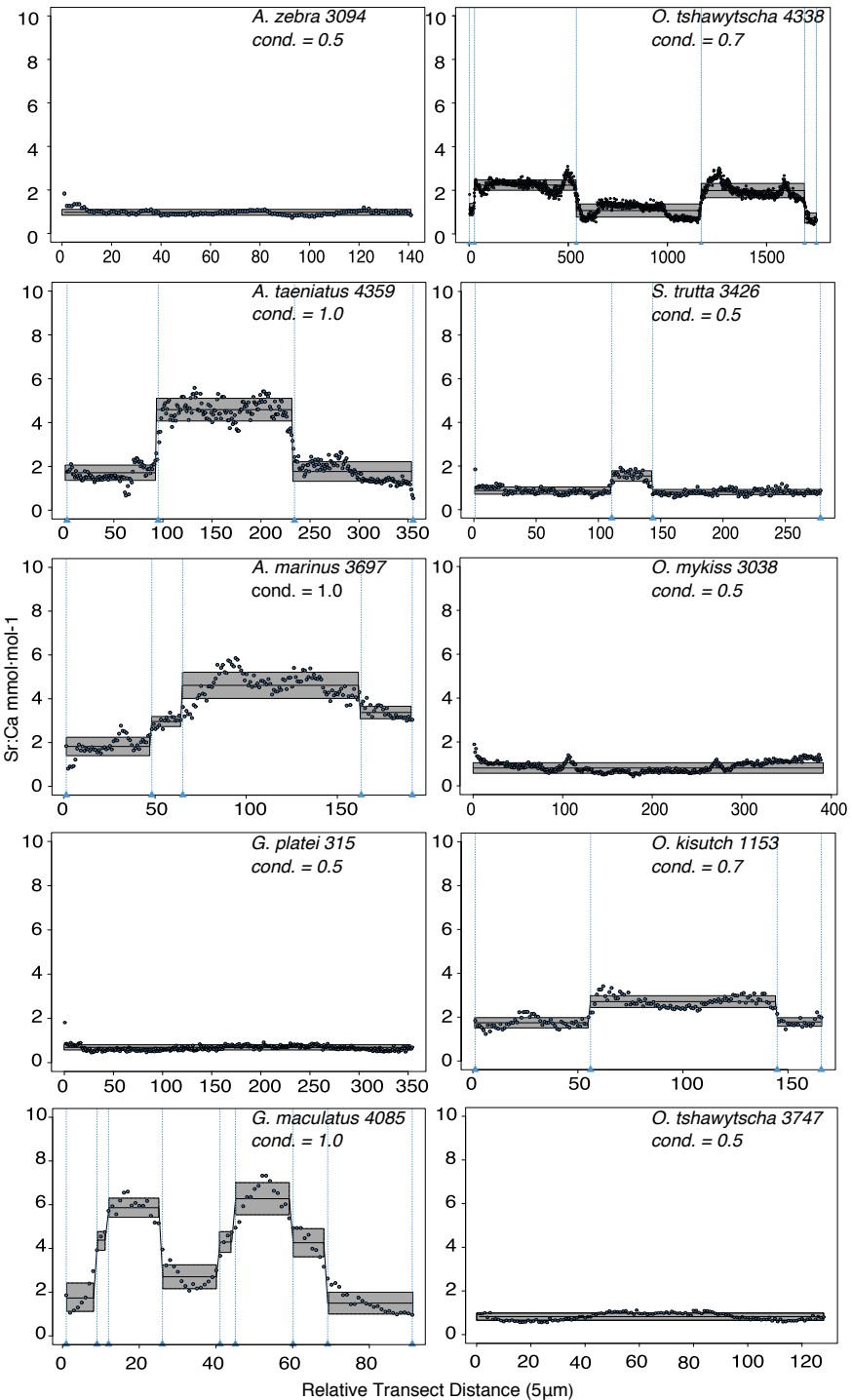

**Figure 4 Detection of discontinuities by semi-supervised CART employed on Sr:Ca ratios for representative individuals of native and exotic fishes in southern Chile.** Numbers after taxonomic names refer to the individual ID of each fish. The mean for each cluster is represented by a continuous black line delimited by a grey box as standard deviation. Vertical dashed lines indicate splitting points induced by the condition used to fit the regression trees, which is reported for each individual graph as "cond".

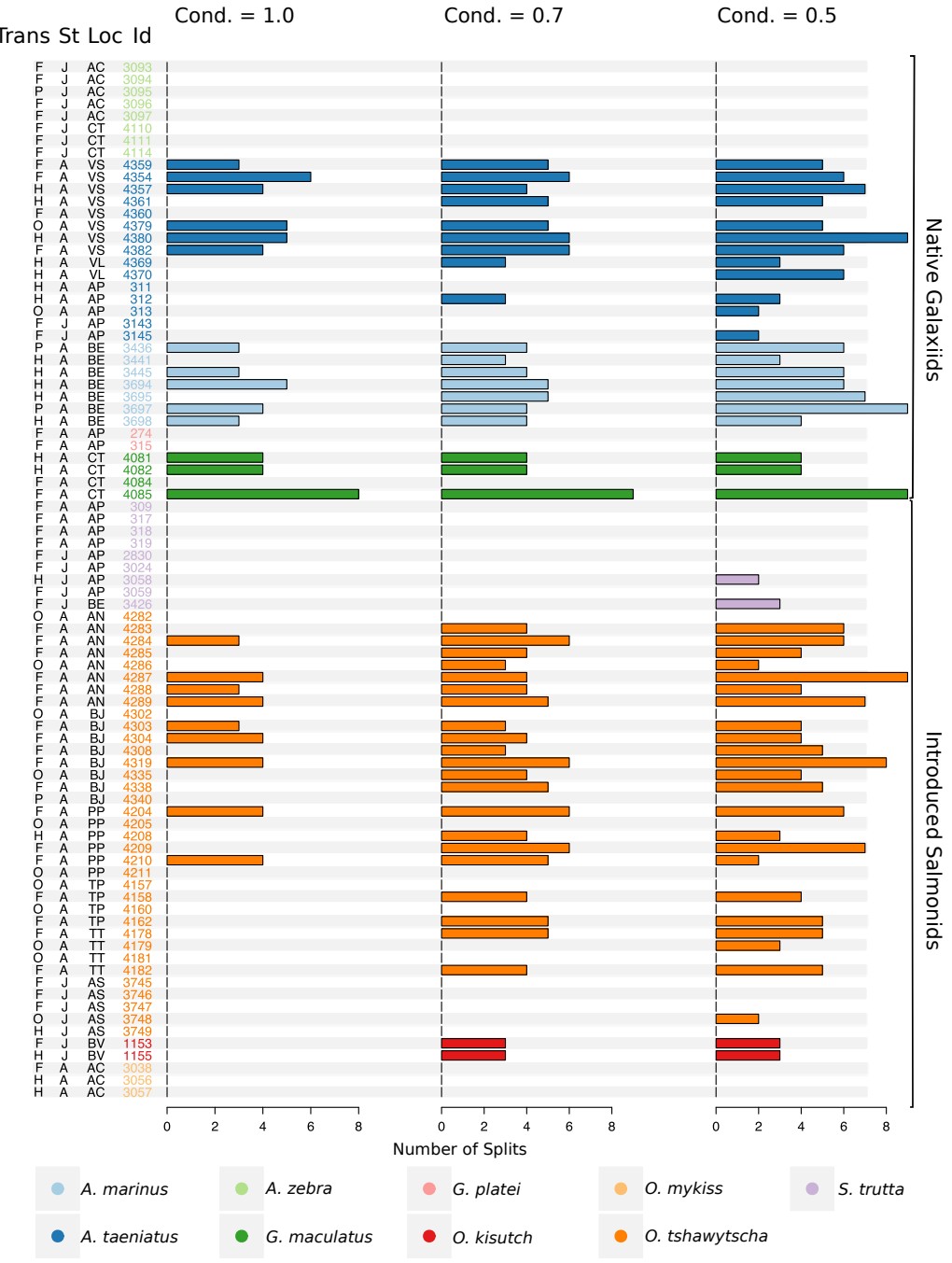

**Figure 5** **Total number of splits obtained by semisupervised CART on univariate Sr:Ca otolith data for all the species included in the study.** Frequency of the total number of splits obtained with different conditions (Cond) by semisupervised CART on univariate Sr:Ca otolith data for all the species included in the study. Original data for native *A. taeniatus, A. marinus, and G. maculatus* and introduced *O. tshawytscha* was divided in more than one homogenous cluster by semisupervised regression trees and led to rejection of the hypothesis of freshwater residency. Details for each individual and split reported in this graph are available in Table S2. (continued on next page...)

**Figure 5 (…continued)**
"Trans" refers to the quality of the otolith transect, that is: "F" is a full or good quality transect, edge –core –edge; "H" is a half transect, edge –core; "P" is a partial transect, edge –core –extra data without reaching the next otolith edge; "O" is a flagged transect which failed to go through the core and may have some missing data. "St" refers to each fish's ontogenetic phase at the time of capture, where "J" is for juveniles and "A" for adult specimens; "Loc" indicates the sampling locality where AC: Aysén-Caro, CT: Chiloé-Tocoihue, VS: Valdivia-Santo Domingo, VL: Valdivia-Lingue, AP: Aysén-Palos, BE: Baker-estuary, AN: Aysén-Ñireguao, BJ: Baker-Jaramillo, PP: Petrohué-Patos, TP: Toltén-Peuco, TT: Toltén-Truful, AS: Aysén-Simpson, BV: Baker-Vargas; "Id" is the unique identification of each fish.

across the otolith transect (Table S2), suggesting that this individual did not drift out to the ocean during its larval stage.

Only one specimen from the low-elevation coastal Palos Lake was assayed for *G. platei*, and results indicate freshwater residency as revealed by uniformly low Sr:Ca ratios across the entire otolith transect (Figs. 3 and 4).

### Introduced salmonids

This study supports established anadromy in *O. tshawytscha* in Patagonia, as previously shown by other authors (*Ciancio et al., 2005*; *Correa & Gross, 2008*; *Arismendi & Soto, 2012*; *Araya et al., 2014*). Data are consistent with changes in Sr:Ca concentration levels that suggest hatching in freshwater, migration to areas with marine influence followed by a return to inland, freshwater areas to spawn (Fig. 4). *O. tshawytscha* specimens collected from the Simpson River do not show as much variation as other *O. tshawytscha* from this study most likely because these fish were all juveniles that had not yet migrated.

The two parr *O. kisutch* analysed revealed one substantial Sr:Ca shift between birth and time of capture. Both otolith profiles showed relatively high Sr:Ca signatures around the core that diminished towards the edges (Fig. 4). These specimens where caught during the summer, about 55 km upstream of the Baker River's mainstream. The observed pattern is consistent with maternal effects imprinting a marine signature near the core, which is formed during the yolk absorption phase (*Kalish, 1990*; *Volk et al., 2000*; *Zimmerman & Reeves, 2002*). Our results add to the sparse documentation of the establishment of self-sustaining *O. kisutch* populations in southern Patagonia (*Górski et al., 2016*). Conversely, *O. mykiss* exhibited a pattern consistent with freshwater residency and minor Sr:Ca fluctuations over the entire life cycle (Figs. 3 and 5).

Evidence of at least two different life cycle patterns emerged for *S. trutta* specimens caught at three different locations. The Sr:Ca transect of juveniles from Lake Caro and adults from Lake Palos showed a pattern consistent with continuous residency in freshwater (Figs. 3 and 5) whereas *S. trutta* from Baker River showed higher values at the primordium (Fig. 4).

### DISCUSSION

This study quantitatively identified significant transitions across otolith profiles using regression trees on Sr:Ca ratios. Native galaxiids showed considerable variation in putative habitat shifts when compared across species, with some species exhibiting differences at the population and individual levels, indicating a high degree of plasticity. Of five

native galaxiids examined, evidence was found for one catadromous (*A. marinus*) and two facultatively amphidromous or catadromous species (*G. maculatus* and *A. taeniatus*). Nonnative salmonids have established populations with a broad array of migratory life histories, reflective of those found in their native ranges. Patterns consistent with anadromy were present in three (*O. tshawytscha, O. kisutch, S. trutta*) of four species included in this study.

Several species appear to regularly use habitats with different levels of marine influence. Otolith profiles that showed variation under the most restrictive analytical conditions were those most likely to exhibit large-scale habitat shifts between different environments and salinity levels (*A. taeniatus, A. marinus, G. maculatus, O. tshawytscha*). Otolith profiles that varied under less stringent conditions hinted at subtler shifts within habitat types.

Results suggest a preponderance of euryhaline (i.e., broad tolerance to different salinity levels) and facultative diadromous species among native galaxiids and introduced salmonids in Patagonia. The inferences on species migratory status by population, reported in Table 1, suggest that some species display a diverse range of life history strategies (facultative diadromy), coinciding with an increasing number of studies reporting flexibility in diadromous patterns for several fishes (*Hicks, Closs & Swearer, 2010*; *Augspurger, Warburton & Closs, 2015*; *Górski et al., 2015*). These studies included Southern Hemisphere fishes and revealed the variability of resident/migratory life histories within species. This work reinforces the shift from the classical view that tended to categorize species as exclusively resident or migratory.

Otolith microchemistry for introduced salmonids suggested that some species have established movement strategies similar to those in their native ranges. The successful establishment of anadromous exotic salmonids in Chile supports the hypothesis that the biotic and abiotic conditions required for diadromy to be maintained (*Gross, Coleman & McDowall, 1988*) are present in Chilean waters.

Current knowledge of preferred habitats and common life-histories in native continental fishes of Chile indicates that approximately 30% of the species have a broad halohabitat distribution, including species adapted to life in saltpans, estuaries and a variety of different diadromous strategies (Table S1). On the other hand, only about 9% of fishes around the world are considered euryhaline and just a few display some form of diadromous migrations (*Nelson, 2006*; *Schultz & McCormick, 2013*). Estuaries of temperate regions often stimulate the evolution of adaptation from seawater to freshwater and can be seen as hotspots for transitions (i.e., regions with a high number of species with propensity for movement between saltwater and freshwater) (*Schultz & McCormick, 2013*; *Ruiz-Jarabo et al., 2016*).

The likelihood of freshwater colonization is affected by multiple factors. Colonization probability increases when habitats are disturbed by extreme events like glaciation, drought, volcanic and anthropogenic activities. Extreme events can leave freshwater habitats with low levels of biodiversity and prone to infiltration and acclimation from migratory or coastal euryhaline fishes. Habitats with wide temporal or spatial ranges in salinity also provide good hotspots for transitions because the acclimation response depends on the physiology of the fish and the timescale of fluctuations. Lower temperatures are prevalent

at mid latitudes and these conditions favor higher rates of colonization, since salinity and temperature interact to affect energetic demands, ion uptake rate and membrane permeability of fishes (*Norman & Greenwood, 1975* as cited in *Bamber & Henderson, 1988*; *McDowall, 1988*; *Lee & Bell, 1999*; *Schultz & McCormick, 2013*).

Specific physiological adaptations can also facilitate freshwater colonization propensity in fishes. For example, specialized gills allow movement between saltwater and freshwater (*Lee & Bell, 1999*; *Schultz & McCormick, 2013*) (i.e., atherinids). Further, certain groups of diadromous fishes (e.g., galaxiids, salmonids) demonstrate plasticity in osmoregulation. Diadromous species migrate between saltwater and freshwater habitats at different stages of their life-cycles, but as shown in this study, some switch to a resident strategy. Euryhalinity and diadromy can therefore be seen as key innovations that enable lineages to radiate into new environments (*Bamber & Henderson, 1988*; *Lee & Bell, 1999*; *McDowall, 2001*; *Schultz & McCormick, 2013*).

Diadromous migrations mostly promote gene flow among populations, but can also lead to landlocking, isolation and cladogenesis (*McDowall, 2001*). Habitat characteristics of coastal zones composed of estuaries separated by long stretches of open shore, may facilitate isolation and landlocking of diadromous and euryhaline fishes into more stable habitats. For example, very high within-species genetic diversity was found in *G. maculatus* across its New Zealand range and along most of the Chilean coast except between 30°S and 42°S in Chile, where *G. maculatus* displayed lower genetic diversity and higher levels of genetic structure (*Lee & Bell, 1999*; *Waters, Dijkstra & Wallis, 2000*; *González-Wevar et al., 2015*). Higher levels of genetic structure found within populations of *G. maculatus* in Chile were hypothesized to depend on glaciation history, but variation in coastal configurations and oceanographic regimes can also be linked to population differentiation and isolation.

Comparative otolith microchemistry analysis suggested that fishes of southern Chile may require a heterogeneous and spatially connected environment to complete their life-cycles. These temperate areas may be considered a favourable environment for the development and maintenance of migratory strategies in fishes and could provide useful tools to evaluate the influence of habitat discontinuities and biogeography on the spatial distribution, colonization rates and genetic diversity of euryhaline and diadromous fishes alike (i.e., atherinids, galaxiids, salmonids). Examining recent freshwater invasions can yield insights into the osmoregulatory systems that enable the invasion of freshwater habitats and offer excellent systems for observing evolutionary adaptation in progress (*Lee & Bell, 1999*; *González-Wevar et al., 2015*).

### Limitations and further studies

Some otolith results may have been influenced by maternal effects or induced by local temporal variation in water chemistry. Further studies may indicate whether the higher Sr:Ca ratios in primordia observed in some species could be attributable to maternal effects or other causes. For example, although the mechanisms are not completely understood, physiological constraints in early ontogeny could increase the rate of Sr absorption into the calcium carbonate matrix of the otolith (*De Pontual et al., 2003*). Also, as the Baker river system is influenced by a large ice field (Campo de Hielo Norte), high amounts of

glacier flour (suspended solids) can contribute to increased salinity levels in water that flows into the estuary (*Vargas, Aguayo & Torres, 2011*; *Marín et al., 2013*). These seasonal salinity changes may promote the uptake of Sr into the otolith matrix and confound the assumption of low Sr in freshwater environments (*Zimmerman, 2005*). Therefore, even though Sr has been traditionally recognized as a very robust marker to discriminate between marine and freshwater environments, several recent studies have indicated that factors such as species-specific variation, environmentally-mediated physiological processes, individual variation and the interaction of different environmental factors can influence Sr uptake into the otolith matrix (*Elsdon & Gillanders, 2002*; *Elsdon & Gillanders, 2004*; *Gillanders, 2005*; *Gillanders et al., 2015*; *Sturrock et al., 2015*).

This study suggests that considerable variation in migratory life history may exist in Chilean fishes, but its inferential scope is restricted by the limited number of samples, which were collected for other research purposes (see *Correa & Hendry, 2012*; *Correa, Bravo & Hendry, 2012*; *Alò et al., 2013*), and by lack of water chemistry samples. The interpretations given for diadromous and resident patterns are limited to the observed Sr:Ca shifts in the otolith profiles and should be considered carefully, especially when referring to the extent of movement between different habitats. Species-specific reference values for Sr:Ca ratios and a more comprehensive sampling will be needed to quantify the extent of fish movements within Chilean continental waters in more detail. Ideally, a variety of different field techniques (natural markers such as stable isotopes, otoliths, statoliths for lampreys, or scales; as well as molecular markers, tagging, trapping and tracking) and laboratory methods (e.g., movement physiology, swimming performance, metabolism) (see *Dingle, 2014*) should be used to characterize daily and seasonal migration patterns. Incorporation of this knowledge could improve design and operation of fish passage structures to benefit native fishes (*Laborde et al., 2016*). Currently, information on fish life history for native fishes in Chile and elsewhere in the temperate Southern Hemisphere is lacking, and fish passage design is generally based solely on professional judgment (*Wilkes et al., 2018*).

## Conservation issues

This study could help refine the conservation priorities for freshwater fishes in southern Chile. Given high endemism and the likelihood of dependence on diadromous behaviour, potential threats to fishes from fragmentation of river-to-estuary networks are correspondingly high. Hydroelectric power development causes loss of hydrological connectivity and alteration of the river flow regime, disproportionately affecting fishes with migratory life histories (*Fullerton et al., 2010*). Comparative otolith microchemistry results underscore the variation in life history strategies that should be accounted for when planning to manipulate water-flow for hydroelectric developments. Diadromous species depend on the habitat diversity and complexity created by unobstructed watersheds and are locally extirpated when barriers preclude movement to essential habitat. Additionally, anthropogenic barriers and alterations to water flow (e.g., hydropeaking) may also negatively affect landlocked populations because such structures disrupt successful reproduction, recruitment and habitat quality (*Alò & Turner, 2005*; *Garcia et al., 2011*). Current hydropower capacity in Chile (~6.000 megawatts of energy connected to the

central grid) is expected to increase (∼11.000 megawatts by the year 2020) building approximately 900 additional hydroelectric power plants due to legislative privileges given to hydroelectric investors such as (i) water allocation rights favouring in-stream productive uses; (ii) ease for hydropower investors to acquire riverside land; (iii) specific tax easements and deferments for hydroelectric investors; and iv) taxes imposed for "ecological" in-stream uses (*Prieto & Bauer, 2012*; *Santana et al., 2014*; *Toledo, 2014*). Preliminary studies have identified most of the hydroelectric potential in the south-eastern sector of the country, in the sub-basins with high elevations and discharge (*Santana et al., 2014*). Development is slated in basins that harbour the majority of native fish species diversity.

Ongoing spread of exotic species threatens native species through negative interactions including predation, competition, behavioural inhibition and homogenization (*Correa & Gross, 2008*; *Penaluna, Arismendi & Soto, 2009*; *Correa & Hendry, 2012*; *Correa, Bravo & Hendry, 2012*; *Habit et al., 2012*; *Arismendi et al., 2014*; *Vargas, Arismendi & Gomez-Uchida, 2015*). In particular, establishment of migration runs of *O. tshawytscha* and *O. kisutch* could trigger additional threats such as the shift of significant amounts of marine-derived nutrients to previously oligotrophic environments (*Helfield & Naiman, 2001*; *Arismendi & Soto, 2012*) and increased competition for limited resources with the native diadromous species. Non-native salmon and trout are also likely to be negatively affected by future hydroelectric dams. Additional hydropower development will almost certainly impact a flourishing tourism industry supported by salmonid recreational fisheries (*Arismendi & Nahuelhual, 2007*; *Vigliano, Alonso & Aquaculture, 2007*).

The evolutionary processes that allowed dispersal and colonization of Patagonian fishes are influenced by the region's unique geography, climate, and geological processes. To ensure conservation of native freshwater, diadromous, and commercially relevant sport fisheries, managers will have to carefully designate and protect critical habitats, and in many cases mitigate obstruction of river flows imposed by dams with appropriate fish passage structures (*Wilkes, Mckenzie & Webb, 2018*). Long-term monitoring should also be a priority to understand the broad impacts of hydropower development on aquatic biodiversity.

## ACKNOWLEDGEMENTS

We are thankful to AP Bravo, IY Quinteros, M Soto-Gamboa, and L Caputo for assistance in the field, and to ML Guillemin for providing field equipment. S Platania, S Barkalow and M Brandenburg assisted with laboratory procedures and logistics. The Museum of Southwestern Biology at UNM provided curatorial assistance. Laboratorio de Dendrocronología y Cambio Global at UACh allowed use of its microscopy facility. We thank P Marquet for revising an earlier version of this manuscript, A Castillo for help with GIS mapping, E Habit for a useful discussion, the anonymous reviewers and PeerJ editor MA Esteban for comments that significantly helped to improve this manuscript.

### Funding

Otolith analysis was funded by a RAC grant from the University of New Mexico, USA. The Government of Chile supported the drafting of this document with a CONICYT Doctoral Fellowship No 21150634 to Dominique Alò in 2015 and to Cristian Correa through grants CONICYT-PAI NO 82130009, and FONDECYT-Iniciación en la Investigación NO 11150990. The funders had no role in study design, data collection and analysis, decision to publish, or preparation of the manuscript.

### Grant Disclosures

The following grant information was disclosed by the authors:
RAC grant from the University of New Mexico, USA.
CONICYT Doctoral Fellowship: No 21150634.
CONICYT-PAI: No 82130009.
FONDECYT-Iniciación en la Investigación: No 11150990.

### Competing Interests

The authors declare there are no competing interests.

### Author Contributions

- Dominique Alò conceived and designed the experiments, analyzed the data, prepared figures and/or tables, authored or reviewed drafts of the paper, approved the final draft, contributed otolith samples.
- Cristian Correa authored or reviewed drafts of the paper, approved the final draft, contributed otolith samples.
- Horacio Samaniego analyzed the data, approved the final draft.
- Corey A. Krabbenhoft performed the experiments, authored or reviewed drafts of the paper, approved the final draft.
- Thomas F. Turner conceived and designed the experiments, contributed reagents/materials/analysis tools, authored or reviewed drafts of the paper, approved the final draft.

### Animal Ethics

The following information was supplied relating to ethical approvals (i.e., approving body and any reference numbers):

The McGill University Animal Care Committee (UACC) Animal Use Protocol No. 5291 approved use and handling of animals.

### Field Study Permissions

The following information was supplied relating to field study approvals (i.e., approving body and any reference numbers):

Specimens were collected under permits No. 3587, 29 December 2006, No. 2886, 4 November 2008 (amendment No. 602, 12 February 2009) and No. 2458, 10 August 2010 provided by the Subsecretaría de Pesca (Chilean Subsecretary of Fishing).

## Data Availability

Supplemental figures are available at figshare: Aló, Dominique; Correa, Cristian; Samaniego, Horacio; Krabbenhoft, Corey; Turner, Thomas (2018): Supplemental Figures. Otolith microchemistry and diadromy in Patagonian river fishes. PeerJ (In press). figshare. Figure. https://doi.org/10.6084/m9.figshare.6387665.v2.

## Supplemental Information

Supplemental information for this article can be found online at http://dx.doi.org/10.7717/peerj.6149#supplemental-information.

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
