# Peer review of "Otolith microchemistry and diadromy in Patagonian river fishes"

_PeerJ, doi:10.7717/peerj.6149_

## Round 0.1 · original submission · Major Revisions

There are some major concerns regarding this manuscript. The objectives are not clearly established, the methodologies have to be deeply described and it is necessary to do a more extensive review of the topic in order to improve the discussion. I invite you to improve your manuscripts according to the suggestions given.

Reviewer 1 ·

Basic reporting

Very good in general. Some suggestions below in order of importance:

Line 22 – ‘…a number of diadromous fishes’. This is ambiguous as it could refer to a number of individuals or of species. Please clarify. Note that in other areas of the manuscript this ambiguity is not an issue.

Line 315-317 – In fact, we are increasingly realising that laboratory experimental approaches to characterising swimming performance are fraught with insurmountable issues regarding fish maintenance, behaviour and scale. Indeed, the species used in Laborde et al. (2016) were not selected on the basis of the need to design fish passage structures, but instead on the basis that they were the only species that would cooperate by surviving and swimming in the laboratory! Unfortunately, the best paper to substantiate this is currently with the publisher after acceptance, so I am unable to refer to it here. But, essentially, the most robust basis we currently have for designing fish passage structures in Chile and elsewhere in the temperate Southern Hemisphere is experience – see the models presented in Wilkes et al. (2018). Wilkes, M., Baumgartner, L., Boys, C., Silva, L.G., O'Connor, J., Jones, M., Stuart, I., Habit, E., Link, O. and Webb, J.A., Fish‐Net: Probabilistic models for fishway planning, design and monitoring to support environmentally sustainable hydropower. Fish and Fisheries.

Line 76 – More specifically, the galaxliids are distributed across the temperate Southern Hemisphere (McDowall, 2002). McDowall, R.M., 2002. Accumulating evidence for a dispersal biogeography of southern cool temperate freshwater fishes. Journal of Biogeography, 29(2), pp.207-219.

Figure 2 – This is a very useful illustration. However, there is no x axis label on the plot. This may be an issue with pdf conversion.

Figures 3, 4 & 5 – The scales of axes are at a much smaller font than axis labels in these plots. Please make them the same size.

Line 98 – ‘…from 6 locations…’ – Unless otherwise specified by the journal, it is convention to spell out numbers <10, i.e. ‘…from six locations…’.

Life 177 – Should read ‘affected the confidence of our interpretations’.

Line 184-186 – I find the way of listing these regimes a little awkward. Could the authors please provide them as bullet points?

Line 297 – I suspect few readers will understand what the ‘Baker corridor’ is. Please provide a definition or use different language. For example, do you mean the valley of the Baker River?

Experimental design

The authors use robust and well-established methods. I have just one suggestion - I am not very familiar with the laboratory procedure involved in analysing otolith annuli but it strikes me that the authors should refer to a standard source(s) or a recent paper(s) using the method in the text around line 107. This will demonstrate clearly that a well-established procedure was used, which from my readings on the procedure appears to be the case here.

Validity of the findings

No comment

Additional comments

I foundd this to be a well-written and enjoyable paper to read. The methods used are robust and well-established and the Supplemental Files are comprehensive and relevant. The ethical statement is sufficient and the appropriate study permits are in place. There are a few, relatively minor ways I could suggest to improve the manuscript below.

Reviewer 2 ·

Basic reporting

The Introduction describe very weakly the reason of this study. How was the choose of fish?, why the author have exotic and native fish as models?.
Used the Hydropower development in Chile, but how is the policy in your country?.

Experimental design

This Ms do not has an experimental design clear, the authors used samples collected for another reason. Neither the fish species chooses for this study.

It can be a good thing to apply the rule of 3R, but in this case is not clear.
The statistical and method are ok.

Validity of the findings

The Otoliths results are interesting, but the movement or diadromy of Salmonid are clear, please to check the Carl Schreck or Steve McCormick papers, and I was checking about Chilean Native species and quickly I found 3 paper the same group:
1.-Isolation Driven Divergence in Osmoregulation in Galaxias maculatus (Jenyns,1848)(Actinopterygii: Osmeriformes), Ignacio Ruiz-Jarabo PLOs ONE 2016
2.-Contrasting Genetic Structure and Diversity of Galaxias maculatus (Jenyns, 1848) Along the Chilean Coast: Stock Identification for Fishery Management Claudio González-Wevar, Journal of Heredity, 2015, 439–447
3.- Phylogeography in Galaxias maculatus (Jenyns,1848) a long Two Biogeographical Provinces in the Chilean Coast. Claudio A.González-Wevar PLOs ONE 2015

1.-This articles you can see that G. maculatus is a diadromy fish, with different osmoregulation response.

I think that authors need to improve the references revision and to include more studies.

Additional comments

Dear Author
Your Ms is interesting but need to improve the introduction, discussion and to improve the literature revision.
The Introduction describe very weakly the reason of this study. How was the choose of fish?, why the author have exotic and native fish as models?.
Used the Hydropower development in Chile, but how is the political in your country?.
The objective is not clear and method need to be more clear, Which is the real reason to do this study, or to they want to take advantage of the samples taken but without a prior reason.
Material and Method
Again, the fish choose, to mixture native vs exotic fish is interesting but here is not clear.
Discussion
Here the author to include the "Limitations and further studies" (line 291) I'm thinking that this paragraph is not necessary and it is better to include in the introduction-material and method. Also, to include "Conservation issues" (line 319) for what? to save an exotic fish? for this reviewer it is not clear.

References revision, I think that here the author need to do a better revision than the show us in this MS version, I was checking about Chilean Native species and quickly I found 3 paper the same group:
1.-Isolation Driven Divergence in Osmoregulation in Galaxias maculatus (Jenyns,1848)(Actinopterygii: Osmeriformes), Ignacio Ruiz-Jarabo PLOs ONE 2016
2.-Contrasting Genetic Structure and Diversity of Galaxias maculatus (Jenyns, 1848) Along the Chilean Coast: Stock Identification for Fishery Management Claudio González-Wevar, Journal of Heredity, 2015, 439–447
3.- Phylogeography in Galaxias maculatus (Jenyns,1848) a long Two Biogeographical Provinces in the Chilean Coast. Claudio A.González-Wevar PLOs ONE 2015

1.-This articles you can see that G. maculatus is a diadromy fish, with different osmoregulation response.
I recommend to submitted this Ms at another journal as J.Fish Biology or Environmental Biology of Fishes, can be more suitable than Peer J.
I hope my suggestions are useful to improve your ms

---

## Round 0.2 · Minor Revisions

Authors have done a very good job improving the original version of the manuscript. Some minor changes are still needed (see reviewer 1 comments).

Reviewer 1 ·

Basic reporting

I was happy to re-review this manuscript. The authors have done a good job of responding to two reviewers’ comments. I can only suggest a few minor corrections. Once these are made, I believe the paper is suitable for acceptance.

Line 82-84: “Differing chemical composition of the otolith from the primordium (core) to the edge hint the different environments in which a fish has lived and allows for hypothesis tests related to patterns of fish movement.” – “Hint” is a little casual for a journal article, and should be followed by “at” in this context. More appropriate would be: “…is indicative of the different environments…”

Line 99-100: “This study sought to typify the potential diadromous characteristics from a wider diversity of native and exotic fish species in Patagonia…” – “Typify” is not quite right here. I suggest “characterize”.

Line 170 – “Phyton scripts” should be “Python scripts”

Line 275 – “self-sustaining coho salmon populations” – You should give the scientific name in parentheses as this is the only time you refer to this species by its common name. Alternatively, just refer to the scientific name.

Experimental design

No further comment

Validity of the findings

No further comment

Additional comments

No further comment

Reviewer 2 ·

Basic reporting

Dear Author
Your Ms is interesting and after the revision, I can see that the MS is improved and for me I don' t have more revisions or comments.

Experimental design

It is ok for me after revision.

Validity of the findings

After the revision the Ms had improved.

Additional comments

Dear Author
Your Ms is interesting and after the revision, I can see that the MS is improved and for me I don' t have more revisions or comments.

---

## Round 0.3 · accepted · Accept

Thank you for improving your manuscript taken into account all the suggestions given.

#